# Effects and Mechanisms of Phthalates’ Action on Reproductive Processes and Reproductive Health: A Literature Review

**DOI:** 10.3390/ijerph17186811

**Published:** 2020-09-18

**Authors:** Henrieta Hlisníková, Ida Petrovičová, Branislav Kolena, Miroslava Šidlovská, Alexander Sirotkin

**Affiliations:** Department of Zoology and Anthropology, Faculty of Natural Sciences, Constantine the Philosopher University in Nitra, 949 74 Nitra, Slovakia; ipetrovicova@ukf.sk (I.P.); bkolena@ukf.sk (B.K.); msidlovska@ukf.sk (M.Š.); asirotkin@ukf.sk (A.S.)

**Keywords:** phthalate, endocrine disruptor, reproductive system, hormone, nuclear receptor

## Abstract

The production of plastic products, which requires phthalate plasticizers, has resulted in the problems for human health, especially that of reproductive health. Phthalate exposure can induce reproductive disorders at various regulatory levels. The aim of this review was to compile the evidence concerning the association between phthalates and reproductive diseases, phthalates-induced reproductive disorders, and their possible endocrine and intracellular mechanisms. Phthalates may induce alterations in puberty, the development of testicular dysgenesis syndrome, cancer, and fertility disorders in both males and females. At the hormonal level, phthalates can modify the release of hypothalamic, pituitary, and peripheral hormones. At the intracellular level, phthalates can interfere with nuclear receptors, membrane receptors, intracellular signaling pathways, and modulate gene expression associated with reproduction. To understand and to treat the adverse effects of phthalates on human health, it is essential to expand the current knowledge concerning their mechanism of action in the organism.

## 1. Introduction

Phthalates are ubiquitous chemicals produced in high volumes. They are used as plasticizers in consumer products. Their widespread application and exposure have raised concerns about human health. Research has shown that exposure to phthalates is associated with various disorders, but most significantly with reproductive disorders [1]. There is a worldwide trend towards increasing reproductive disorders, such as hormone-dependent cancers, infertility, and decreased fecundity. In 2015, 8–12% of couples worldwide were infertile or had decreased fecundity [2], and 6.7% of females were infertile [3]. Over the last 50 years, the 32.5% decrease in sperm concentration in men has been remarkable. One of the causes of current reproductive disorders could be environmental chemicals, such as phthalates [4]. It is essential to address the problem of an increase in fertility disorders. Characteristics of the mechanisms of phthalates’ effects are essential to solve this growing problem. However, in the present, there is not enough of this intricate knowledge. Therefore, we consider it necessary to create such a comprehensive overview of the topic.

In this review, we attempted to present an overview of the current knowledge concerning the phthalates’ effect on reproductive health at multiple levels. This review integrates the results from in silico, in vitro, in vivo studies, and epidemiological studies to show the complexity of phthalates’ effects on reproduction. We summarized the knowledge of general reproductive regulators, phthalate toxicity, and phthalates’ effect on male and female reproduction at the clinical, hormonal, and intracellular levels.

## 2. Regulators of Reproduction

The development and functions of the reproductive system are under the control of numerous genes. Hormones produced by the hypothalamic–pituitary–gonadal (HPG) axis regulate their expression [5].

### 2.1. Hypothalamic–Pituitary–Gonadal (HPG) Axis

Hypothalamic neurons produce neural signals, as well as neurohormones-kisspeptin, leptin, and others in the arcuate nucleus and periventricular region [6]. They control the synthesis of gonadotropin-releasing hormone (GnRH). GnRH released from the hypothalamic preoptic area stimulates the anterior pituitary GnRH receptors (GnRHRs). These receptors promote the secretion of anterior pituitary gonadotropins luteinizing hormone (LH) and follicle-stimulating hormone (FSH) via the mitogen-activated protein kinase (MAPK) and cyclic adenosine monophosphate (cAMP) signaling pathways [7,8]. FSH triggers the growth and development of ovarian follicles in the ovaries and sperm production in the testis. FSH and LH stimulate the secretion of sex steroids and protein hormones, such as anti-Müllerian hormone and insulin-like peptide 3, as well [9]. Anti-Müllerian hormone is essential for the inhibition of Müllerian duct development in males [10] and the inhibition of multiple ovarian follicles’ development in females [11]. Insulin-like peptide 3 is vital for testicle descent [12]. The production and function of steroid hormones will be discussed in the next chapter.

The HPG axis regulates the levels of reproductive hormones via a positive and negative feedback loop. Higher levels of hormones from gonads inhibit the secretion of GnRH from the hypothalamus and gonadotropins from the adenohypophysis. The positive feedback loop is needed before ovulation when high levels of estradiol stimulate LH secretion from the adenohypophysis [9]. One of the main functions of the HPG axis is the regulation of steroidogenesis. The next chapter is focused on the characterization of the steroidogenic process.

### 2.2. Steroidogenesis

The first step of the process of steroidogenesis is cholesterol synthesis. Cholesterol presents the precursor molecule for all steroidogenic hormones, including sex hormones. The essential enzymes for steroidogenesis include P450 and HSD enzymes from mitochondria and endoplasmic reticulum. These enzymes can convert steroidal precursors to final hormones such as testosterone, dihydrotestosterone, progesterone, and estradiol [13].

LH binds to LH receptor (LHR) of thecal cells (in case of ovarian follicles) and Leydig cells (in case of testis). FSH binds to the FSH receptor (FSHR) of granulosa cells in the ovarian follicles. In these cells, FSH and LH activate a broad spectrum of signaling pathways leading to the gene expression of steroidogenic enzymes and the conversion of steroids to final products [7,8].

Testosterone and dihydrotestosterone are leading male masculinization hormones. Testosterone is essential for the morphological differentiation of the internal genital organs and maintaining spermatogenesis. Dihydrotestosterone is an androgen 10-fold more potent than testosterone and is associated with the differentiation of male external genitals and male secondary sexual characteristics. However, androgens are vital for follicle maturation in females. Estradiol and progesterone are essential hormones for postnatal female reproductive system development and to enable ovarian and menstrual cycle, pregnancy, and labor. Estradiol is needed to maintain spermatogenesis in males, as well [9]. In general, steroid hormones have proliferative features. They are crucial for germ cell proliferation leading to sperm production and ovarian follicles’ development [14,15]. Moreover, sex steroids induce cell proliferation in the non-reproductive tissue, such as the bladder [16], precursors of myotubes [17], or neural stem cells [18]. Furthermore, sex steroids have anti-apoptotic properties. Data show that after castration and during hormone insufficiency, the apoptotic process occurs. This process occurs mainly in the hormone-dependent tissue, such as the prostate gland, uterus, oviduct, and mammary gland [19]. Steroid hormones need to be transported with transport proteins in the bloodstream to reach this hormone-dependent tissue [20]. Thus, the next chapter is focused on the characterization of the most significant transport protein for sex steroids.

### 2.3. Sex Hormone-Binding Protein (SHBG)

Most of the sex steroid hormones are transported in an inactive bind with transport proteins, such as SHBG. The relative binding affinity of some sex steroids for SHBG is as follows from the highest affinity: dihydrotestosterone, testosterone, androstenediol, estradiol, and estrone. SHBG levels elevate when testosterone decreases, and when estradiol increases [21]. SHBG transports steroids to target tissue where they can act by numerous mechanisms. The next chapter is focused on the definitions of the primary mechanisms of steroid action.

### 2.4. Mechanisms of Steroid Action

The main steroid effect on the cells is mediated by genomic action [22]. However, there is also a known rapid non-genomic mechanism of action by activating membrane receptors and protein kinases in signaling pathways [23] and by epigenetic processes as well [24].

#### 2.4.1. Genomic Action of Steroid Hormones

The genomic mechanism of steroid action is mediated through nuclear receptors (NRs). There are two main groups of nuclear NRs—type I and II. Type I or steroid hormone receptors include mineralocorticoid receptor, glucocorticoid receptor, androgen receptor (AR) [25], progesterone receptor (PR) with three isoforms (PR-A, PR-B, PR-C) [26], and two types of estrogen receptor (ERα and ERβ) [27]. Type II or non-steroid hormone receptors include thyroid hormone receptor, vitamin D receptor, retinoic acid receptor, peroxisome proliferator-activated receptor (PPAR), and others [25]. Between orphan receptors—receptors with an unknown ligand can be found estrogen-related receptors (ERRα, ERRβ, ERRγ) [28].

NR signaling pathway consists of specific steps as follows. The ligand penetrates through the cytoplasmic membrane and binds to the ligand-binding domain (LBD) of NR in the cytoplasm or nucleus. Before the ligand binds to the NR, the receptor is coupled with heat-shock proteins (HSPs) in an inactive state. When the ligand binds the receptor, the activation and dimerization of this complex occur. The ligand and receptor without HSPs translocate to the nucleus. They bind to the hormone response elements (HREs), which is a DNA sequence in specific gene promoters. Gene expression starts when coactivators bind to the promoter sequences [25,29].

#### 2.4.2. Non-Genomic Action of Steroid Hormones

Steroid hormones coupled with NRs can activate cytoplasmic protein kinases associated with various signaling pathways—cyclic adenosine monophosphate (cAMP), calcium, MAPK, nuclear factor kappa B (NF-κB), and phosphoinositide 3-kinase (PI3k/Akt). In some cases, steroids may bind to membrane receptors, e.g., cytokine receptors, G protein-coupled receptors (GPCRs) that are not primarily designed for steroid binding [23,30,31,32].

In other cases, steroids bind to membrane receptors designed for steroid hormones as well, such as membrane androgen receptors (GPCR6 and ZIP) [33], membrane estrogen receptors (mERα, mERβ, GPER, ER-X, ERx and Gq-mER) [34] and membrane progesterone receptors (mPRα-ε) [35]. GPCR6 and ZIP9 are membrane androgen receptors regulating Sertoli cell function in males by non-classical testosterone signaling, moreover, they are also associated with prostate and breast cancer onset [33]. GPER together with ER regulate Leydig cell function in males [36]. However, GPER is associated with reproductive tissue cancer, such as breast, ovary, endometrium, testis or prostate [37]. The membrane progesterone receptor regulates oocyte maturation, labor, and sperm motility and reproductive organs cancer onset [35]. Membrane steroid receptors mediate the steroid effect mainly by cellular proliferation, apoptosis, and metabolic functions, as described in the following.

The cAMP pathway is mainly stimulated by GPCRs. It regulates critical physiological processes including metabolism, secretion, calcium homeostasis, muscle contraction, and gene transcription [38]. Calcium belongs to the signal molecules, which regulate a broad spectrum of cell processes, such as oocyte activation [39] and sperm capacitation during fertilization [40], myosin ATP-ase activation during muscle contraction [41], the release of neurotransmitters to synapsis [42] or the regulation of the apoptotic process associated with calpains, which are proteases dependent on the levels of Ca2+ [43]. The PI3k/Akt signaling pathway is involved in the regulation of apoptosis and cell division. Akt activates several target proteins involved in cell determination, metabolism, and protein synthesis [44]. MAPKs are serine–threonine kinases regulating various cellular processes associated with cell determination. Receptors for cytokines and growth factors, such as transforming growth factor (TGF), stimulate the MAPK signaling pathway. Most of the signaling pathways interconnected with steroid signaling are associated with the metabolism and regulation of cell proliferation and cell death [45]. NF-κB creates a group of transcription factors that are involved in various biological processes, including immune response, inflammation, cell growth, survival, and development [46].

These intracellular pathways can mediate steroid hormones action on cell proliferation (cellular multiplication, which leads to the growth of cell population) and of apoptosis (genetically programmed cell death). Two different pathways stimulate apoptosis. Intrinsic pathway (or mitochondrial pathway) of apoptosis occurs through the activation of caspases—enzymes catalyzing the cleavage of the cell proteins. The external pathway is activated by an external “death” signal bound to the tumor necrosis factor receptor, which activates caspases [47]. Steroids, but also some peptide hormones and other factors, regulate cell proliferation and apoptosis (see Table 1).

#### 2.4.3. Epigenetic Processes

Epigenetics presents any process that influences gene expression without changing the DNA sequence and leads to modifications that can be inherited. DNA methylation, histone modifications, and non-coding RNAs synthesis belong to mechanisms of epigenetics [50]. Epigenetics can influence the expression of genes associated with reproductive system differentiation and functioning [24]. In the case of the reproductive system, the epigenetic process can regulate genes controlled by NRs, e.g., the genes involved in the control of steroidogenesis, steroid degradation, and reproductive functioning. This regulation controls the degree of hormonal stimulation in the target tissue [51].

In conclusion, the HPG axis regulates hormonal secretion in the organism. The main products of the HPG axis are steroid hormones. Transport proteins, such as SHBG, transport steroid hormones to the target tissue. Steroid hormones regulate processes in the target tissue by genomic, non-genomic, and epigenetic mechanisms. Regulators of reproduction are often the targets of phthalates’ action. The next chapter is focused on the primary mechanism of phthalate toxicity, physical and chemical properties of phthalates, and the definition of phthalate exposure sources.

## 3. Structure, Source, and Toxicity of Phthalates

Phthalates are esters of 1,2-benzene dicarboxylic acid. Their structure varies depending on the number of side chains, which are formed by dialkyl, alkyl, or aryl groups [52]. They are colorless or slightly yellowish, oily, odorless substances, very slightly soluble in water [53]. Their solubility decreases with the prolonging chain [54]. Phthalic acid derivatives are much more readily soluble in organic solvents. The longer the side chain, the higher their fat solubility, and the higher the boiling point is present [53].

Phthalates are ubiquitous chemicals used in industrial manufacturing as plasticizers, supplying plastic products with elasticity [55]. Phthalates are divided into two main groups based on their molecular weight. Long-chain or high molecular weight phthalates (HMWP)—di(2-ethylhexyl) phthalate (DEHP), di-iso-nonyl phthalate (DiNP), di-iso-decyl phthalate (DiDP), di-n-octyl phthalate (DnOP), di(2-propylheptyl) phthalate (DPhP) are used as a part of polyvinyl chloride (PVC). Short-chain or low molecular weight phthalates (LMWP)—dimethyl phthalate (DMP), diethyl phthalate (DEP), benzylbutyl phthalate (BBzP), di-n-butyl phthalate (DnBP) and di-iso-butyl phthalate (DiBP) [56] are applied in the manufacture of personal care products, solvents or adhesives [57]. The environment is contaminated with phthalates. Phthalates were found in samples of soil (0.03–1280 mg/kg), in drinking water samples (0.16–170 µg/dm^3^), in samples of air (<0.4–65 ng/m^3^), and in dust samples (2.38–4.1 g/kg) [58]. Humans can be exposed to phthalates via different ways—via food intake, by inhalation, intravenously, and through dermal contact [56,57]. Dermal absorption is a more critical type of exposure for LMWP and ingestion for HMWP [54]. Moreover, they are capable of transplacental transition, and therefore they can exert their toxic effects within embryonic and fetal development [59]. Therefore, phthalates can alter the development of reproductive systems [60]. A critical developmental window for reproductive system development is during gonadogenesis (5th–18th week of gestation). However, the functional maturation of the reproductive system lasts until adolescence [61]. Phthalates as EDs can impair the development of the genital system during prenatal as well as the postnatal period of ontogenesis [60].

Phthalate metabolism consists of two parts of biotransformation: hydrolysis and conjugation [62]. Dialkyl phthalates are metabolized to monoalkyls by enzymes [60] that exhibit lipase and esterase activity [63]. Through hydrolysis, diesters become more bioactive monoesters [64] with an average half-life of 12 h [65]. In Table 2 are listed the primary and secondary metabolites of selected phthalate diesters [66]. The reference dose of phthalate intake in humans is 20 µg/(kg/bw/day) and the tolerable daily intake is 50 µg/(kg/bw/day) [58].

Phthalates belong to the group of endocrine disruptors (EDs), which affect the hormonal balance of the organism. They can alter the development and function of the hormone-dependent structures of the reproductive system [67]. Humans and animals are exposed to mixtures of EDs at low doses in the environment. These chemicals interact with each other via different mechanisms, which can lead to synergistic, additive, or antagonistic toxic health effects [68,69]. Phthalates, like hormones, exert their physiological effects in low doses rather than in high doses. This phenomenon is called non-monotonic toxicity [70].

To conclude, phthalates are man-made chemicals used in the plastic industry. There are several ways to be exposed to phthalates, mainly via inhalation, ingestion, and transplacental transition. Phthalates belong to the chemicals known as EDs. They modulate the hormonal balance of the matured organism as well as the developing organism. This modulation of hormonal balance has a significant impact on the reproductive health of males.

## 4. Phthalates’ Action on Male Reproductive Health

Sufficient data are pointing to the phthalate’s effect on male reproductive health mostly during the prenatal period but also during postnatal ontogenesis [1]. This effect will be discussed in the next chapters.

### 4.1. Phthalates Can Influence Testicular Function

Phthalates interfere with male reproductive system development. The primary targets of the phthalates’ action on the testis are Sertoli cells and Leydig cells [71]. In vitro studies demonstrated that DEHP exposure at 40, 80 and 160 μM and dibutyl phthalate (DBP) exposure at 10 and 100 mg/L caused the apoptosis of TM3 Leydig cells and Sertoli cells of Male Sprague-Dawley rats, respectively [72,73]. Animal studies showed that prenatal exposure to DiNP at 100 mg/kg resulted in multinucleated gonocytes among male Sprague-Dawley rats [74]; prenatal exposure to DBP at 500 mg/kg was linked with malformed seminiferous tubules among male Wistar rats [75]; prenatal exposure to dicyclohexyl phthalate (DCHP) was associated with multinucleated gonocytes (at 100 and 500 mg/kg), focal testis dysgenesis (at 500 mg/kg), abnormal Leydig cells morphology and abnormal aggregation (at 10, 100, 500 mg/kg) [76]. Postnatal DBP exposure at 200, 400 and 600 mg/kg induced seminiferous cords necrosis and the absence of spermatogenesis in rats [77]. Phthalate exposure induced the impairment of sperm parameters in men. The level of mono-iso-nonyl phthalate (MiBP) was positively associated with sperm motility and negatively associated with total sperm count [78]. Urinary levels of monobutyl phthalate(MBP) and monoethyl phthalate (MEP) were associated with decreased sperm concentration and reduced sperm motility, respectively, in 125 Chinese men visiting an infertility clinic [79]. The semen levels of mono-n-butyl phthalate (MnBP), mono-(2-ethylhexyl) phthalate (MEHP), mono-(-2ethyl-5-hydroxyhexyl) phthalate (MEHHP), mono-(-2ethyl-5-oxohexyl) phthalate (MEOHP) were associated with decreased sperm concentration, MEHHP, MiNP with reduced sperm motility and MnBP with spermatic DNA damage in 344 Polish men visiting an infertility clinic [80]. The semen levels of monobenzyl phthalate (MBzP), MEHP, MEHHP were associated with reduced sperm motility in 1247 Chinese men visiting an infertility clinic [81]. The semen levels of MBzP were linked with an abnormal number of sperm heads and flagella [82]. Moreover, DnBP disturbed the maturation and activation of human sperm before fertilization, which was associated with early motility response and acrosomal exocytosis [82].

In conclusion, the data have shown that phthalates affect testicle function. It leads to a broad spectrum of disorders of male reproduction. One of these disorders is testicular dysgenesis syndrome (TDS), which will be discussed in the next part of the chapter.

### 4.2. Phthalates Can Induce Testicular Dysgenesis Syndrome (TDS)

As mentioned above, prenatal exposure to phthalates disrupts the development and function of Leydig and Sertoli cells. This leads to other disorders, e.g., decreased testis weight, spermatogenesis impairment, and external genital malformations (shortened anogenital distance, hypospadias, and cryptorchidism) [83]. These symptoms in animal models were extensively reported in the review by Sharpe and Skakkebaek [83] as well as in boys and men in the review by Bay et al. [84]. These signs are collectively referred to as phthalate syndrome in animals and TDS in humans. However, in the case of TDS in men, testicular cancer may occur as well [85,86].

To conclude, phthalates induce testicular damage and external genital malformations. These symptoms are known as phthalate syndrome in male rats and TDS in men. Other male reproductive disorders related to phthalate exposure include dysfunctions of puberty. This topic will be discussed in the next part of the chapter.

### 4.3. Phthalates Can Influence Male Puberty and Its Dysfunctions

Pubertal development is a multiphase process controlled by several hormonal mechanisms [87]. Puberty is initiated with the increased secretion of GnRH from the hypothalamus. It stimulates sex steroid production and the development of secondary sexual characteristics. These signals represent the termination of the dormant condition of the HPG axis during childhood [88]. Phthalates affect the male reproductive system not only in utero, but they play a role in male postnatal sexual development and puberty. Whether it is the acceleration or delay of puberty onset, it depends on the timing of phthalate exposure, on their concentration, and other factors [89]. In vivo studies observed the biphasic effect of DEHP action on the onset of puberty. Exposure to DEHP at 750 mg/kg caused a delay in pubertal onset, and the DEHP at 10 mg/kg caused precocious puberty in male rats [90]. Epidemiological studies showed conflicting results. Shi et al. [91] noticed that the urinary levels of MnBP, MEHHP, MEOHP were associated with a delay in pubertal onset and levels of MnBP were negatively associated with testicular volume in 252 Chinese boys at the age of 7–14 years. A Chinese case-control study reported significantly higher urinary levels of MnBP and MEP in boys with the constitutional delay of growth and puberty (n = 60) compared to the control group (n = 120) at the age of 8–15 years [92]. Moreover, they found that the urinary levels of MnBP and MEP were negatively associated with decreased testosterone [92]. Furthermore, a Chinese epidemiological study involving 222 boys at the age of 6–14 years, reported the negative association between the urinary levels of MnBP and both, pubertal onset and a growth of pubic hair [93]. In contrast, the results of the Danish study involving 84 boys at the age of 6–13 years showed that the urinary levels of MnBP were associated with precocious pubertal onset [94]. A study by Berger et al. [95] noticed the positive association between the maternal urinary levels of MBzP, monocarboxy-isononyl phthalate (MCNP), monocarboxyoctyl phthalate (MCOP), mono-(3-carboxypropyl) phthalate (MCPP), ΣDEHP and pubertal onset in 159 US boys (most of them were Hispanics) at the age of 9–13 years. This association was stronger in obese boys [92]. In the study by Mieritz et al. [96], they did not observe significant correlations between phthalate metabolites in the urine samples of 555 boys at the age of 16–20 years.

We hypothesize that the conflicting results in the associations between phthalate exposure and the onset of puberty are likely attributed to ethnicity. Ethnic population groups have slightly different physiology values, e.g., hormone levels or puberty onset. The lowest levels of androgens [97] and a later onset of puberty are observed in the group of Asian boys [98]. The highest levels of androgens [99] and an earlier onset of puberty are observed in African American boys [100]. There are also differences between ethnic groups in the xenobiotic metabolism, such as different the isoforms of enzymes involved in oxidation, hydrolysis and conjugation. These changes result in the altered activity of metabolic enzymes [101,102]. Therefore, there can be ethnic changes in phthalate metabolite concentrations. For instance, higher levels of phthalate metabolites were observed in non-Hispanic blacks and Hispanics in comparison with non-Hispanic whites and Asians [103]. However, there might not only be genetic differences between ethnicities, but environmental ones as well. Humans from different regions can be exposed to different concentrations of phthalates in their environment. For example, levels of phthalates in indoor dust samples were lowest in the samples from North America (500.02 μg/g), next from Europe (580.12 μg/g) and the highest concentrations were from Asia (945.45 μg/g) [104]. Moreover, there can be ethnic differences in the use of cosmetic products or consumer practices causing exposure to different mixtures of EDs [103].

We can hypothesize that the effect of phthalates on puberty onset in various ethnic groups can be different based on the altered physiological values, metabolic processes in xenobiotic biotransformation and exposure to different mixtures of EDs.

In conclusion, phthalate exposure is associated with the alternation of puberty in boys. Phthalates accelerate or delay the onset of pubertal signs in different ethnic groups of boys. However, the direct effect of phthalate exposure on puberty onset in different ethnicity is not clear yet. Another male reproductive disorder related to phthalate exposure is a cancer of male reproductive organs. This topic will be discussed in the next part of the chapter.

### 4.4. Phthalates Can Induce Cancer in Male Reproductive Organs

Even if the organ developmental programming occurs in utero, the aftereffects of prenatal environmental exposure may manifest during adulthood. It is known as the “fetal basis of adult disease”. An example of the fetal basis of adult disease is cancer [105], including testicular and prostate cancer. Cell proliferation and apoptosis are natural processes which are strictly regulated by tumor suppressor genes. Any alternation in the functioning of these genes leads to uncontrolled cell division, which is known as tumorigenesis [106].

Prostate gland development is defined by their cell proliferation differentiation and apoptosis, which in turn depend on the testosterone exposure and testosterone/estradiol ratio. Phthalates, acting like anti-androgens, disrupt the hormonal balance between estradiol and testosterone, leading to abnormal prostate cell proliferation [107,108]. An in vitro study demonstrated the stimulatory effect of phthalates DEHP (at 10^−4^–10^−6^ mol/L), BBzP and DBP (at 10^−6^–10^−7^ mol/L) on the proliferation of PC-3 human prostatic carcinoma cells [109]. The same study reported the stimulatory effect of DEHP, BBzP (at 10^−7^–10^−10^ mol/L) and DBP (10^−8^–10^−10^ mol/L) on the proliferation of 22RV1 human prostatic carcinoma cells [109]. Zhu et al. [110] showed a positive association between the exposure to BBzP at 10^−6^–10^−7^ mol/L and the proliferation of LNCaP and PC-3 human prostatic carcinoma cells. An in vivo study noticed that the prenatal exposure to DnBP at 100 mg/kg from gestational day 12 to postnatal day 21 was linked with the proliferation of the prostatic cells in male Wistar rats [108].

Epidemiological studies are insufficient. The study by Chang et al. [107] observed a positive association between the levels of DEHP metabolites in urine samples and benign prostate hyperplasia and prostatic enlargement in 207 elderly men from Taiwan. There is no study to prove the direct effect of phthalate exposure on prostate cancer development.

There is still a lack of direct evidence of the associations between prenatal exposure to EDs and testicular cancer occurrence [111,112]. One of the high risks of testicular cancer occurrence is cryptorchidism. Cryptorchidism is a disorder of testicle descent, which is frequently associated with prenatal exposure to environmental pollutants, such as phthalates [85,111]. Newborns with cryptorchidism are at a 30–50 times increased risk of developing testicular cancer then newborns with descended testis [113].

To our knowledge, there is lack of studies showing the direct effect of phthalate occupational exposure on testicular cancer. Therefore, we analyzed the relations between occupational exposure to mixtures of EDs in PVC and their potential effect on testicular cancer occurrence in male workers. Epidemiological studies did not find a clear association between PVC exposure and testicular cancer occurrence [114,115]. The Swedish case-control study involving 981 men with testicular cancer and 981 control subjects aged 20–75 showed no significant relationship between occupational exposure to PVC and testicular cancer [116]. The study of Hardell, Ohlson, and Fredrikson [117] included 148 men with testicular cancer and 315 controls aged 30–75. This study showed that the exposure to PVC increased testicular cancer risk in men [117]. The US case-control study included 527 mothers of sons with testicular cancer and 562 mothers of controls aged 20–29. This study showed that the maternal use of cosmetics products during pregnancy, such as facial lotion more than once a week, was associated with a higher risk of testicular cancer occurrence in their sons diagnosed approximately at the age of 20–29 years [118].

EDs may induce the cancer of the prostate gland and testis. However, there is no direct evidence of the phthalates’ effect on testicular and prostate cancer occurrence in the epidemiological studies.

Taken together, the available data have demonstrated that prenatal as well as postnatal phthalate exposure is associated with male reproductive disorders, such as TDS, the modulation of pubertal onset, and the manifestation of pubertal signs. However, further studies are required to examine the relationship between phthalate exposure and prostate and testicular cancer occurrence because there is a lack of studies providing direct evidence of this relationship. Phthalates harm both male and female reproductive health. However, the phthalates’ effect on female reproductive health has not been studied so extensively as the male reproductive toxicity of phthalates. Phthalate impact on female reproduction will be discussed in the next chapter.

## 5. Phthalates’ Action on Female Reproductive Health

Data point to the phthalates’ effect on female reproductive health, mostly during the prenatal period but also during postnatal ontogenesis. This effect will be discussed in the next chapters.

### 5.1. Phthalates Can Influence Ovarian Function

In vivo and in vitro studies have pointed to the endocrine-disrupting properties of phthalates in females. These studies observed phthalates’ effects on ovarian function in female rats and mice. Phthalates modified the follicular development by the inhibition of antral follicles development and decreased the number of antral follicles in mice (mixture of EDs—DEHP, DBP, BBzP, and two alkylphenols at 1 mg/kg and 10 mg/kg) [119], the stimulation of follicle development in mice (MEHP at 500 and 1000 mg/kg) [120], and a decreased number of follicles in mice (DEHP at 200 and 500 g/kg) [121]. Moreover, the exposure to DEHP at environmentally relevant doses affected oocyte growth, maturation and ovulation in females of *Danio rerio* [122]. In *Caenorhabditis elegans*, the exposure to DEHP at 10 mg/L decreased the number of oocytes and induced DNA damage in oocytes [123]. In vitro experiments showed that oocytes from DEHP-exposed female mice at 20 and 40 μg/kg/day induced defects in oocyte meiosis [124]. Then, an in vitro experiment showed similar results, with the exposure to 10 and 100 μM of DEHP inhibiting meiotic progression in mice [125]. Oocyte meiosis was altered in mice female progeny (DEHP at 20 μg/kg/day) [126]. Zhang et al. [127] showed that in vitro exposure to 10 and 100 µM of DEHP decreased germ cell nest breakdown in newborn mouse ovaries. A review by Zhang et al. [128] summarized the effects of DEHP exposure on oogenesis and folliculogenesis. Data showed that DEHP induced the altered development of the primordial germ cells, germ cell survival, meiotic progression and increased follicle atresia [128]. In addition, DEHP disturbed the maturation and activation of oocytes before fertilization via meiotic maturation inhibition and oxidative stress [129]. Exposure to DEHP at 25 mg/m^3^ by inhalation [130] and prenatal exposure to a mixture of DEP, DEHP, DnBP, DiNP, DiBP, and BBzP at 20, 200 and 500 mg/kg [131] impaired the estral cycle, particularly ovulation and estradiol synthesis in rats and mice, as well [130,131]. The imbalance of the hypothalamic–pituitary–ovarian axis furthermore negatively affected the development and function of the reproductive system of female progeny [132]. These effects of exposure to the mixture of phthalates were observed in the second and third generation of the progeny of mice [131]. The similar results occurred in the progeny of *Caenorhabditis elegans* after the prenatal exposure to DEHP at 20 mg/L [133]. Therefore, we can assume that phthalate exposure has a transgenerational as well as a multigenerational effect on fertility in female animal models [131].

In conclusion, the data have shown that phthalates affect ovarian functions leading to full-spectrum disorders associated with reproduction. Moreover, the impairment of female reproductive health can have a transgenerational and multigenerational effect. One of these disorders is premature ovarian failure, which will be discussed in the next part of the chapter.

### 5.2. Phthalates Can Induce Premature Ovarian Failure (POF)

POF in women is the condition when ovarian function terminates before 40 years of age. POF consists of symptoms such as amenorrhea, increased gonadotropins levels, and decreased estradiol [134]. In vivo studies noticed that the exposure to DEHP and mixture of EDs induced POF in mice [119,120]. Epidemiological studies noticed consistent results in women. The anti-estrogenic activity of phthalates acted by inhibiting estradiol production in the ovary, leading to anovulation and premature ovarian insufficiency [135,136]. Higher DEHP exposure was associated with a higher risk of decreased ovarian reserve in 215 women visiting the Fertility Center at the Massachusetts General Hospital, USA [137]. The study by Gallicchio et al. [135] observed that the occupational exposure to EDs led to a five times higher risk of POF in hairdressers compared with control subjects limited to Caucasian women only. The possible mechanism of phthalates’ action on developing POF is that DEHP can increase the FSH level [138], which is associated with a high rate of follicle growth and subsequent premature ovarian depletion [139].

To conclude, the results from in vivo and epidemiological studies have shown that the exposure to some phthalates is associated with decreased levels of estradiol, decreased ovarian reserve, and anovulation. This is collectively referred to as POF. Other female reproductive disorders related to phthalate exposure are dysfunctions of puberty. This topic will be discussed in the next part of the chapter.

### 5.3. Phthalates Can Induce Dysfunctions of Female Puberty

Phthalates can alter the onset of puberty in female experimental animals. In vivo studies have observed inconsistent results. Studies of Moyer and Hixon [120] and Patiño-García et al. [119] found that MEHP at 500 and 1000 mg/kg [120] and a mixture of phthalates at 1 and 10 mg/kg [119] induced delayed puberty onset in female mice. Exposure to DEHP at 5 and 25 mg/m^3^ by inhalation [130] and oral exposure to DEHP at 1000 mg/kg [140] sped up the onset of puberty in female rats in the studies of Ma et al. [130] and Liu et al. [139]. Phthalate exposure can modulate the onset of puberty in girls. Srilanchakon et al. [141] suggested that levels of MEP were higher in Thai girls with precocious puberty (n = 42) than in control group (n = 77). Hashemipour et al. [142] observed a similar pattern in Iranian girls aged 7–10. ∑DEHP concentrations were higher in girls with precocious puberty (n = 87) in comparison with control subjects (n = 63) [142]. The US multiethnic longitudinal study consisting of 1151 girls aged 6–8, observed a positive association between ∑HMWP and pubarche (pubic hair development) [143]. A Chinese epidemiological study involving 208 girls aged 6–14 reported the negative association between the urinary levels of MEHP, MEHHP, MEOHP and both breast development and the initiation of the menstrual cycle [93]. Similarly, Binder et al. [144] noticed a positive association between the urinary levels of ∑DEHP and the initiation of the menstrual cycle in 200 girls aged 6–9 in Chile. Higher levels of phthalates were observed in girls with higher body mass index (BMI) [144]. Phthalate exposure can cause precocious puberty in girls acting like obesogens [145]. A higher percentage of body fat causes higher levels of leptin, which stimulates the HPG axis to induce puberty [9]. The Taiwanese case-control study involving 71 girls with central precocious puberty (CPP) and 29 girls in control group aged showed significantly higher levels of phthalate metabolites and kisspeptin-54 in girls with CPP in comparison with the control group; the results showed a positive correlation between the urinary levels of MnBP and levels of kisspeptin-54, as well [88]. Significant secretion of kisspeptin is associated with the development of puberty and luteinizing hormone-releasing hormone I secretion [88]. On the contrary, the study by Kasper-Sonnenberg et al. [146] noticed that higher levels of MEP, MnBP, MBzP, and ∑DEHP were associated with delayed puberty onset. In a Danish study, Frederiksen et al. [147] noticed significantly lower levels of ∑MBP and ∑DEHP in girls with precocious puberty (n = 24) in comparison with age-matched healthy girls (n = 184) at the age of 7.4–9.9 years. They observed a negative association between ∑MBP, ∑all phthalate metabolites and pubic hair development in healthy girls and girls with precocious puberty (n = 725). In the longitudinal study by Berger et al. [95] involving 179 US girls aged 9–13 (most of them were Hispanics), it was noticed that the urinary levels of MCNP, MCOP and MCPP were negatively associated with pubic hair development among normal weight girls. The levels of MCNP, MCOP, MCPP and ∑DEHP were negatively associated with the initiation of the menstrual cycle among normal weight girls and levels of ∑DEHP and MBzP were associated with breast development in all girls. A longitudinal multiethnic study included 1239 girls from the USA aged 6–8 years at proband recruitment. Urinary concentrations of ∑DEHP and MBzP were negatively associated with pubic hair and breast development, respectively, during 7 years of the study duration, mostly among normal weight girls [89].

In conclusion, phthalate exposure can be associated with the alternation of puberty in girls. Phthalates accelerate pubertal onset in girls with a higher BMI. It can be associated with the obesogenic effect of phthalates. In contrast, phthalates induce the delayed onset of puberty in some studies. The reason for the inconsistent link between phthalate exposure and puberty onset in girls is not clear yet. Another female reproductive disorder related to phthalate exposure is a dysfunction of pregnancy. This topic will be discussed in the next part of the chapter.

### 5.4. Phthalates Can Induce Dysfunctions of Pregnancy

Phthalates affect the length and process of pregnancy. In vivo studies observed that DEHP exposure at 250 and 500 mg/kg [148], at 50 and 200 mg/kg [149], and at 20, 200 and 500 mg/kg [130] during the pregnancy in mice, inhibited placental angiogenesis [148,149] and induced miscarriage and obstructed labor in next generations, respectively [130]. Results from in vivo studies are consistent with epidemiological studies. Toft et al. [150] and Messerlian et al. [151] noticed that a high exposure to DEHP was associated with the spontaneous abortion of 303 pregnancies from a Fertility Center in the USA [151] and 430 pregnancies from Denmark [150]. A similar pattern was observed in the case-control studies of Yi et al. [152] and Liao et al. [153]. A study of Yi et al. [152] included women aged 22–35 from Shanghai in 150 matched pairs of case-controls. This study reported significantly higher levels of monomethyl phthalate (MMP) and MEHP among the women with pregnancy loss [152]. A study by Liao et al. [153] involving women aged 20–49 from Taiwan were divided into case (n = 103) and control groups (n = 76). This study reported significantly higher levels of ∑DBP among the women with recurrent miscarriage [153]. For the late stage of pregnancy, increased levels of ∑DEHP had a protective effect against miscarriage [150]. A US study by Adibi et al. among 283 pregnant women showed that urinary levels of ∑DEHP during pregnancy caused birth after the 41^st^ gestational week or increased the probability of the section [154]. Then, a US nested case-control study by Ferguson et al. [155] involved 130 pregnancies with preterm birth and 352 controls from Brigham and Women’s Hospital, Boston, Massachusetts. This study reported the opposite phenomenon: the exposure to DEHP was associated with a risk of preterm birth. Similarly, Latini et al. [156] conducted a study involving 84 newborns (39 males and 45 females) with an average gestational age of 38.4 ± 2.2 weeks. This study observed that higher levels of MEHP in cord blood were associated with decreased gestational age at delivery. The reason for the conflicting results is that phthalates and their metabolites, such as DEHP and mono-(2-ethylhexyl) phthalate MEHP, can modulate both PPAR [157,158] and prostaglandins [153,154]. PPARs are necessary for maintaining a pregnancy. DEHP and its metabolites could bind to PPAR and prevent the maternal–fetal communication which allows birth to be initiated [157,158]. Prostaglandins are signaling molecules, which induce contractions of the uterus, leading to birth or abortion. DEHP stimulated the secretion of prostaglandins, which could induce spontaneous abortion or preterm birth [156,159].

To conclude, phthalates induce dysfunctions of pregnancy, such as prolonged pregnancy or a shortening of pregnancy and miscarriage by the modulation of PPAR and prostaglandin activity. Phthalates have an impact not only on pregnancy outcomes but also on the onset of cancer in female reproductive organs. This topic will be discussed in the next part of the chapter.

### 5.5. Phthalates Can Induce Cancer in Female Reproductive Organs

Exposure to phthalates can induce the proliferation of cancer cells, and hence initiates gynecological cancers. This was confirmed by the following studies. Choi et al. [160] observed the stimulatory effect of exposure to DnBP at 10^−5^ M on BG-1 human ovarian cancer cell proliferation. Yang et al. [161] showed that MEHP at 10^−7^–10^−9^ M triggered the proliferation of human cervical cancer cell lines HeLa and SiHa. Epidemiological studies noticed consistent results. A Romanian study involved 37 women diagnosed with cervical cancer aged 26–76. In this study, the urinary levels of MEHP were associated with an increased size of cervical tumors [162]. A Korean case-control study recruited 53 women diagnosed with leiomyoma (35.3 ± 0.8 years) and 33 controls without leiomyoma (32.6 ± 1.4 years). This study showed that the urinary levels of MECPP were higher in women with leiomyoma than in control subjects [163]. A US multiethnic study involved 1227 women aged 25–54, whereby 151 of them were diagnosed with leiomyoma. This study reported significantly higher urinary levels of MBP and lower levels of MEHP in women with leiomyoma [164]. Pollack et al. [165] observed that higher levels of monomethyl phthalate (MMP) were associated with a decreased risk of leiomyomas in 494 women aged 18–44 from the USA. Then, a US multiethnic study included 57 women diagnosed with leiomyoma aged 26–54. This study noticed that the urinary levels of MCNP were positively associated with leiomyoma size [166]. In addition, a US study recruited 3003 women aged 25–85, most of whom were non-Hispanic white, whereby 20 of them were diagnosed with ovarian cancer. In this study, the occurrence of ovarian cancer was positively associated with the urinary levels of MEHHP [167]. A systematic review by Fu et al. [168] involved nine studies with 6579 probands. After the analysis of nine original research articles, this systematic review reported that ΣDEHP was positively associated with a risk of leiomyoma [168].

Some phthalates exhibit estrogenic properties, which increase levels of endogenous estradiol. Higher levels of estradiol stimulate cell proliferation and growth, leading to the onset of hormone-dependent types of cancer, such as ovarian, uterine, and cervical cancers [169,170].

In conclusion, the results from in vitro and epidemiological studies have shown that phthalate exposure is associated with an increased risk of cancer in female reproductive cells and organs.

Taken together, the available data have suggested that phthalate exposure in females can lead to reproductive disorders, such as POF, decreased fecundity, adverse pregnancy outcomes, gynecological cancer, or a modulation of pubertal onset and pubertal symptoms in girls. Altered reproductive health can be influenced by phthalates’ impact on the hormonal system. The next chapter is focused on the hormonal mechanisms of phthalates’ action.

## 6. Hormonal Mechanisms of Phthalates’ Action on Reproductive Functions and Health

Reproductive disorders caused by phthalates are associated with HPG axis dysregulation at different regulatory levels. At the hormonal level, phthalates interact with steroidogenic enzymes and hormones [171] as well as with SHBG [172].

### 6.1. Phthalates’ Effect on the Hypothalamic–Pituitary–Gonadal (HPG) Axis and Steroidogenesis

Phthalates interfere with the regulation of the HPG axis. They alter the levels of GnRH, LH, and FSH. This leads to the disturbing activity of steroidogenic enzymes with the subsequent effects on the steroid hormones. The results from in vitro and in vivo studies are shown in Table 3.

The conflicting results in the associations between phthalate exposure and the hormones of the HPG axis in experimental animals are likely attributed to the use of various types of phthalates, the dose of phthalates, the use of different experimental animals and animal strains and the timing of the exposure to phthalate. Various types of phthalate diesters can induce diverse effects. For instance, reproductive toxicity is related to the side chain length of phthalates. Phthalate diesters with a side chain length of C4-6, such as DEHP, DBP or BBzP, are able to interfere with reproductive health [187]. The dose of phthalates can also be the source of conflict in results. Phthalates, like hormones, exert their physiological effects instead in low as in high doses. This phenomenon is called non-monotonic toxicity [70]. This type of toxicity was shown in various studies [121,178,185]. For instance, Hannon et al. [121], showed that the exposure to DEHP at 10 µg/mL increased the levels of Cyp19a1, Hsd17b1 and exposure to DEHP at 100 µg/mL decreased levels of those enzymes in cultured mouse antral follicles. In contrast, some studies are showing that phthalates induced linear toxicity [176,181]. For example, Ha et al. [181] observed that with the increasing dose of DEHP in Sprague-Dawley rats, levels of testosterone, FSH and LH decreased. Some animal species and strains, seem to be less sensitive to phthalate-induced toxicity, and part of this variability may be attributed to differences in phthalate biotransformation [187,188]. For instance, Martinez-Arguelles et al. [173] and Meltzer et al. [138] observed that in female Sprague-Dawley rats, prenatal DEHP exposure at 300 mg/kg/day induced a decrease in estradiol levels. On the other side, Brehm et al. [178] noticed increased levels of estradiol in female CD mice after the prenatal exposure to DEHP at 20, 500, 750 mg/kg/day. Exposure to phthalates may induce different toxic effects depending on the sex of the animal, as there are intersexual differences in the individual isoforms of biotransformation enzymes. There are specific intersexual differences in the activities of those enzymes as well [189,190]. According to Reposukou et al. [191] the exposure to mixture of phthalates (DBP, BBzP, DEHP, DiNP) at 13 mg/kg/day decreased levels of Cyp17a1 in female C57/BL6 mice, contrarily to female C57/BL6 mice with increased levels of Cyp17a1 at same dose. The timing of the exposure to phthalates can be the source of discrepancies. Prenatal exposure to phthalates may induce more severe effects because pregnancy is a sensitive window for toxicant exposure as a result of fetal development [192]. According to Meltzer et al. [138], the postnatal exposure to DEHP decreased the estradiol levels in female Sprague-Dawley rats. In contrast, in the study by Brehm et al. [178] were noticed increased levels of estradiol in female CD mice after prenatal exposure to DEHP.

In the associations between phthalate exposure and hormones of HPG axis in epidemiological studies can be observed some discrepancies in the levels of testosterone, estradiol and gonadotropins in men and testosterone in women (see Table 4). The conflicting results in testosterone levels can be likely explained by the timing of exposure to phthalates and by reproductive disorders occurrence. Hart et al. [193] showed that prenatal exposure to some phthalates could induce increased levels of total testosterone in 20 year-old men. Similarly, in men with benign prostatic hyperplasia, estradiol levels, testosterone and steroidogenic enzymes aromatase, and 5α-reductase were increased [107]. Other studies observed associations between postnatal exposure to phthalates in healthy and infertile men and decreased testosterone [80,194,195,196,197,198]. Epidemiological studies yielded some conflicting results in the association between phthalate exposure and estradiol levels in men. Mendiola et al. [199] observed decreased estradiol levels in men with increased levels of MEHP. In the studies of Chang et al. [107] and Al-Saleh et al. [197] we observed positive associations between the metabolites of DEHP in men with prostatic hyperplasia and DEP in healthy men, respectively, and levels of estradiol. These conflicting results are likely attributed to the exposure to different phthalate diesters and reproductive disorders’ occurrence. Results from in vitro studies confirm the existence of these inconsistencies. In vitro studies showed that DEHP exposure decreased as well as the increased expression of estrogen receptor [200,201]. The study by Lee et al. [202] confirmed that DEP acts like an estrogenic chemical. The most significant differences in the results of epidemiological studies were observed in the levels of gonadotropins in men. The study by Al-Saleh et al. [197] included 599 men attending an in vitro fertilization clinic in Saudi Arabia, whereby 47.7% of those men were diagnosed with male infertility. The median age of those men was 36.23 years. The data of this study showed a positive relationship between the urinary levels of MEHP and the serum levels of FSH and LH [197]. Similarly, Axelsson et al. [194] observed a positive association between the urinary levels of MHiNP and MoiNP and the serum levels of FSH and LH in 314 healthy Swedish men aged 17–20. A Chinese study involved 1066 potentially infertile men or men of infertile couples with an average age of 29.1 years. This study reported negative associations between urinary levels of MiBP and MnBP and serum levels of FSH and LH [195]. The Swedish study included 234 young healthy men aged 18–21. This study noticed a negative association between the urinary levels of MEP and the serum levels of LH [203]. A US study by Duty et al. [204] showed negative relationships between the levels of MBzP and FSH in 295 healthy men aged 18–54. Based on the results of cohort studies, we could not draw conclusions on the relationship between the phthalate exposure and effects on FSH and LH levels in men. In epidemiological studies with female subjects, we observed conflicting results in the levels of testosterone. Cathey et al. [205] observed different effects of DBP and DEP metabolites on testosterone levels in women. MEP decreased and MHBP increased its levels. These conflicting results between MEP and MHBP are likely attributed to the exposure to different phthalate diesters. In general, phthalates are considered as the anti-androgenic xenobiotics [83]. Interestingly, a Chinese study including 84 healthy men aged 29.8 ± 6.6, showed that the urinary levels of ∑DEHP were positively correlated with testosterone [206].

In conclusion, phthalate exposure interferes with the HPG axis by hormonal and steroidogenic enzyme level fluctuation resulting in the aftereffects on both male and female fertility. However, in these results, conflicting observations can be made. Therefore, further research is needed to explain the direct mechanism of phthalates’ effect on the HPG axis. Phthalates interact not only with hormones and enzymes but with transport proteins as well.

### 6.2. Phthalates’ Effect on Sex Hormone-Binding Globulin (SHBG)

Phthalates act as the active hormonal agents in the organism. They can compete with natural hormones by binding to the transport globulin [172,196]. In silico studies observed that HMWP had a higher affinity to SHBG in comparison with LMWP. For this reason, HMWP can have a possibly more significant effect on hormonal transport inhibition [172]. The in silico study by Sheikh et al. [214] showed that phthalates’ substituent DEHT had a higher affinity to SHBG as well. MEP, monobutyl phthalate (MBP), and MEHP were inversely associated with the levels of SHBG in boys [196,215]. MEHP was inversely associated with SHBG in boys. ∑DEHP was positively associated with the levels of SHBG in girls [196]. These studies support the anti-androgenic and pro-estrogenic effects of DEHP, MEP, and mono-n-butyl phthalate (MnBP) [196,215].

The listed reports represent strong evidence that phthalates modulate the transport process of steroids. Phthalates can compete with steroid hormones for SHBG and therefore, to suppress their binding to SHBG. This can result in the altered bioavailability of steroid hormones. These processes lead to altered levels of the reproductive hormones’ reservoir and HPG axis dysregulation, which can induce reproductive disorders.

Based on the results from in silico, animal, and epidemiological studies, we hypothesized that phthalate exposure interacts with HPG axis activity. The intact HPG axis is essential for proper reproductive development during the prenatal and postnatal periods. If the levels of sex hormones are insufficient or excessive, reproductive disorders may occur. Altered hormonal balance can be influenced by the phthalates’ impact on the intracellular signaling. The next chapter is focused on the mechanisms of intracellular signaling, which are modulated by phthalate exposure.

## 7. Intracellular Signaling Mechanisms of Phthalates’ Action on Reproductive Functions and Health

Reproductive disorders caused by phthalates are associated with HPG axis dysregulation at different regulatory levels. At the intracellular level, phthalates interact via genomic, non-genomic, and epigenetic mechanisms of action to alter gene expression, cell proliferation, and apoptosis, mostly in gonadal cells.

### 7.1. Phthalates’ Effect on Signaling Pathways of Peptide Hormones

Phthalates can affect the HPG axis and steroidogenesis through interaction with genes for GPCRs—receptors for GnRH on pituitary cells, receptors for FSH and LH on Leydig cells, ovarian cells—granulosa and thecal cells. Postnatal exposure to DEHP in female Wistar rats at 3000 [176] and 1000 mg/kg/day [131] increased the expression of Gnrhr [132,176]. A similar pattern was observed in Fshr. Postnatal exposure to DEHP in female Wistar rats at 10 mg/kg/day increased the expression of Fshr [216]. Postnatal exposure to BBzP in male Sprague-Dawley rats at 10 and 100 mg/kg/day increased the expression of Lhr [185]. Similarly, at 100 and 1000 mg/kg/day, the postnatal exposure to BBzP in male Sprague-Dawley rats increased the expression of Fshr [185]. Postnatal exposure to DEHP in female and male CD-1 mice at 0.5 and 5 mg/kg/day decreased the expression of Fshr and Lhr [217]. According to Repouskou et al. [191], postnatal exposure to a mixture of phthalates (DBP, BBzP, DEHP, DiNP) in male C57/BL6 mice at 0.26 mg/kg/day decreased the expression of Lhr. Contrarily, at 13 mg/kg/day, the mixture of phthalates (DBP, BBzP, DEHP, DiNP) increased the expression of Lhr in male C57/BL6 mice [191]. Taken together, the higher exposure to phthalates was associated with an increased expression of Gnrhr, Fshr and Lhr in rats and mice. Contrarily, the exposure to lower levels of phthalates was associated with a decreased expression of Lhr and Fshr in rats and mice. This could further affect steroidogenesis. Phthalates influence not only the expression of peptide receptors but also the activity of nuclear receptors. This will be discussed in the next part of the chapter.

### 7.2. Phthalates’ Effect on Nuclear Receptors (NRs)

The reason why phthalates interfere with steroid hormones lies in a similar chemical structure. Phthalic benzene ring copies the steroid A ring. They are parts of ligands bound to the receptor [218,219]. Therefore, the phthalates can bind to the receptors as the agonists or antagonists. It depends on the size of the side chain [220].

Phthalates influence the activity of AR, ER, and PPAR. Interaction with AR and ER can lead to the impaired action of endogenous signal molecules on hormone-dependent tissue [1,221], such as the endometrium, gonads, breast, adipose tissue, liver, prostate, adrenal gland and skin [222,223,224]. Interaction with PPAR can lead to impaired placental function, which is associated with the impaired timing of baby delivery and spontaneous miscarriage occurrence [157,158].

Phthalates act as agonists or antagonists on the NRs [52]. In silico studies observed a high affinity of some phthalates to NRs: DPhP, MBzP, MEHP, BBzP, mono-(2-propylheptyl) phthalate (MPhP) to hAR [225], DINP, DPhP, BBzP, MPhP, DnOP to hERα; MPhP, MEHP, MnHP to hERβ [226], DINP, DnDP, DEHP, DnOP, BBzP, DPhP, DDP to hPPARα; DiDP, DnDP, mono-iso-decyl phthalate (MIDP), DINP, MEHP to hPPARγ [227]. Moreover, the study by Kambia et al. [228] showed that the metabolites of the phthalate substituent–terephthalate, MEHHT, also had a higher affinity to ER and AR.

Based on in vitro studies, DEP, DEHP, DiBP, DnBP are the anti-androgenic [229,230] and DnBP, DEHP are anti-estrogenic xenobiotics [230,231], which means that they can bind to AR and ER to block the effect of androgens and estrogens on particular cells [229]. Moreover, DEHP can bind to the PPAR and block its effects [231]. Brzozowski et al. [219] and Georget et al. [232] described the antagonistic action on the NR, which is shown in Figure 1. Anti-androgens and anti-estrogens block the conformational change of the NR in the complex with the ligand. The NR is unable to obtain active conformation and dissociate chaperones from this complex, which prevents the transcription initiation. Besides, many of the AR antagonists have sizeable substituents. These substituents prevent the receptor from forming the correct conformation to activate the coactivators and thereby inhibit transcription. This mechanism of action is called active antagonism and is shown in Figure 2 [233]. On the other side, phthalates can stimulate the activity of some NR [228]. ER and PPAR are not ligand-specific receptors. The PPAR, ERα, and ERβ ligand-binding domains are significantly larger, allowing for access to diverse groups of small molecules, especially environmental chemicals [158,234]. DEP, DEHP, DiBP, DiNP, DnBP can significantly activate ER [229,230,231,235], DEHP, DiBP and DnBP can stimulate the activity of PPARs [231,236]. In Table 5 are selected phthalates divided based on their estrogenic/anti-estrogenic and androgenic/anti-androgenic affinity [237].

In conclusion, phthalates modulate the activity of NRs. They act as the agonists and antagonists of ER, AR, and PPAR. Phthalates act through the NRs on the expression of the genes associated mainly with reproductive system development and functioning. This effect will be discussed in the next chapter.

#### Phthalates’ Effect on Gene Expression Mediated by NRs

Phthalates can affect the genes listed in Table 6 in male and female reproductive system tissue [238,239]. NRs regulate the expression of these genes [240]. Altered gene expression directly influences reproductive health and hormone secretion [238,241].

In vitro and animal studies observed an impaired gene expression of steroidogenic enzymes and hormones associated with male reproductive health after the exposure to phthalates [245,246]. Besides, the phthalates modification of gene expression can also influence the onset of cancer. For instance, Yong et al. [247] observed that in the prostate cell lines exposed to MEHP at 1, 5, 10, and 25 μM, the increased levels of Ptch led to abnormal cell proliferation and prostate cancer occurrence. Phthalate exposure may dysregulate the development of the female reproductive system by various mechanisms. Phthalates can influence their function. In the animal studies, maternal exposure to DEHP at 350, 700 mg/kg [238] and 2, 20, 200 mg/kg [239] induced the up-regulation of Wnt4, Foxl2, and Rspo1 in mice embryos, and this could lead to premature ovaries occurrence [238,239].

Via epigenetic mechanisms, phthalates can affect the exposed individual as well as the first and second generation of progeny. This mechanism of toxicity is possible due to the epigenetic modulation of genes in germ cells. Depending on the number of generations affected by epigenetic influence, there are transgenerational or multigenerational effects [248]. Prenatal exposure to 10 and 100 mg/kg DEHP-modulated gene expression and subsequently, hormone activity via DNA methylation in male Wistar rats. This process of hypermethylation occurred in the SF-1 and Sp-1 transcription factors and genes for steroidogenic enzymes of Leydig cells. This could trigger TDS or other reproductive disorders in males [241]. On the contrary, phthalates, e.g., BBzP and DBP at 10-7 M induced DNA hypomethylation or the demethylation of the ERα gene sequences of MCF7 cells [249]. Maternal exposure to DEHP at 40 μg/kg inhibited the DNA methylation of Igf2r and Peg3 genes in F1 and F2 mouse oocytes [250]. These genes are essential for germ cell proliferation. This interaction influenced the quality of the next generation of germ cells [250].

In conclusion, the results from in vitro and in vivo studies have shown that phthalate exposure is associated with altered gene expression and epigenetic processes, which are regulated by NR signaling. This impairment can be observed mainly in gonadal cells or cells related to the reproductive system. Phthalates also affect the cells related to the reproductive system by other mechanisms. The next chapter is focused on the phthalate modulation of cell proliferation and apoptosis within the reproductive system.

### 7.3. Phthalates’ Effects on Apoptosis and Proliferation of Cells of the Reproductive System

Phthalates can inhibit spermatogenesis via apoptosis. The process of spermatogenesis requires a balance of pro-apoptotic and anti-apoptotic signaling to maintain the optimal conditions for sperm maturation [251]. In the study by Giammona et al. [252], MEHP was administered orally to several rodent species and strains (1 g/kg to gld and B6 mice and Sprague-Dawley rats; 2 g/kg to Fisher rats). MEHP activated the external pathway of apoptosis and the NF-κB signaling pathway [252]. The signaling pathway PI3k/Akt in interaction with NF-κB is an essential factor in the first protection of germ cells against apoptosis induced by MEHP [253]. Results from in vitro studies noticed that MEHP at 10 and 50 μM [254], DnBP at 10^−7^–10^−5^ M [201], BBzP at 10^−7^–10^−6^ mol/L [110] influenced the activity of the MAPK signaling pathway to induce apoptosis in C18-4 spermatogonial stem cells [254], and cell proliferation, in BG-1 ovarian cancer cells [201], and human prostate cancer LNCaP and PC-3 cells, respectively [110]. DnBP induced cancer onset via the interaction with ER and the modulation of the MAPK signaling pathway [202]. They observed that the DnBP exposed the LNCaP prostatic carcinoma cells (DnBP dose at 10^−6^–10^−5^ M) activated TGFβ signaling via the MAPK signaling pathway [202]. In ovarian cancer cells, the exposure to DEHP activated ER and cyclin D via the MAPK pathway. The higher activity of cyclin D triggered cell proliferation [201]. Similarly, in prostate cancer cells, the exposure to DEHP, BBzP, and DBP stimulated cell proliferation via ERK5 and p38 [110]. These activities could lead to prostate and ovarian cancer onset [110,201].

To conclude, the results from in vitro studies have shown the phthalates’ effect on apoptosis and the proliferation of cells within the reproductive system. Phthalates modulate the activity of signaling pathways, such as MAPK, NF- κB, and PI3K/Akt, to delay apoptosis and stimulate cell proliferation. Studies show that phthalates interact with protein kinases with subsequent effects on NRs, leading to cancer onset. Phthalate exposure can affect gonadal cells specifically by interaction with their maturation and activation. This mechanism will be discussed in the next chapter.

### 7.4. Phthalates’ Effects on Maturation and Activation of Gonadal Cells before Fertilization

Phthalates can disturb the maturation and activation of gonadal cells before fertilization via the oscillation of Ca2+ intracellular levels. DnBP activated the sperm-specific CatSper channel and increased intracellular Ca2+ levels in vitro. Increased levels of Ca2+ are associated with motility response and acrosomal exocytosis. Exposure to phthalates could lead to the disruption of a chain of actions linked with fertilization [82]. EDs inhibited the meiotic maturation of porcine oocytes in vitro by the alternation of Ca2+ intracellular levels. However, this association was statistically significant only in BPA, not in DEHP and BBzP [255].

Based on the results from in vitro studies, phthalates increase Ca2+ levels in the gonadal cells to inhibit the maturation and activation of these cells. This interaction is associated with the alternation of fertilization.

Taken together, phthalates act at the intracellular level of the reproductive system via interaction with membrane receptors GnRHR, LHR, FSHR, which regulate steroidogenesis. Phthalates increase the levels of Ca2+ levels in the gonadal cells to inhibit the maturation and activation of these cells. Moreover, they interfere with NRs, e.g., AR, ER, PPAR, as the agonists and antagonists. Phthalates alter cell proliferation and apoptosis via crosstalk between MAPK, NF-kB, PI3K/Akt, and NR. This alternation can lead to impaired spermatogenesis and cancer occurrence. Phthalate exposure is associated with altered gene expression and epigenetic processes, which are regulated by NRs.

## 8. Conclusions

The objective of this review was to analyze the phthalates’ effects on the reproductive system and their endocrine and intracellular mechanisms (see Figure 3). This paper represents an extensive review of results from in silico, in vitro, in vivo studies, and epidemiological studies with the focus on human reproductive health.

Phthalates are man-made chemicals used in the plastic industry. There are several ways to be exposed to phthalates, mainly via inhalation, ingestion, and transplacental transition. Phthalates belong to the chemicals known as EDs. They share a similar structure with steroid hormones, which is necessary for their interaction with receptors designed for steroids. This interaction is linked with endocrine disruption. They modulate the hormonal balance of the matured organism as well as the developing organism, due to their ability to pass through the placental barrier. The available data demonstrate that phthalate exposure is associated with male reproductive disorders, such as TDS, the modulation of pubertal onset, and the manifestation of pubertal symptoms. However, further studies are required to examine the relationship between phthalate exposure and prostate and testicular cancer occurrence because there is a lack of studies providing direct evidence of this relationship. Phthalate exposure in females can lead to reproductive disorders, such as POF, decreased fecundity, adverse pregnancy outcomes, gynecological cancer, or the modulation of pubertal onset and pubertal symptoms in girls.

At the hormonal level, phthalates interact with HPG axis activity, which is crucial for proper reproductive development during the prenatal and postnatal periods. If the levels of sex hormones are insufficient or excessive, reproductive disorders may occur.

At the intracellular level, phthalates act via interaction with the signaling of membrane receptors GnRHR, LHR, FSHR, which regulate steroidogenesis. Phthalates modulate levels of Ca2+ levels in the gonadal cells to disrupt the maturation and activation of these cells. Moreover, they can interfere with nuclear receptors, e.g., AR, ER, PPAR, as the agonists and antagonists. Phthalates alter cell proliferation and apoptosis via crosstalk between MAPK, NF-kB, PI3K/Akt, and NR. This alternation can lead to impaired spermatogenesis and cancer occurrence. Phthalate exposure is associated with altered gene expression and epigenetic processes, which are regulated by NRs.

The results of our review suggest that phthalate exposure is associated with reproductive disorders with potential transgenerational or multigenerational effects. The increased use of phthalates and other EDs in the plastic products industry in the last 70 years can explain the worldwide higher prevalence of reproductive disorders.

## 9. Future Directions of Phthalate Research

The analysis of the available literature indicates some inconsistency or deficit of the available knowledge concerning the particular aspects of phthalates’ action on reproduction. For example, there is contradicting information concerning the association between phthalate exposure and pubertal onset in both girls and boys. The available publications report that phthalates can either delay or induce precocious puberty onset. The other example could be a deficit of information about cancer occurrence in phthalate-exposed men. There are only indirect indications of phthalate influence on cancer onset in humans: a higher risk of cancer occurrence was observed in male newborns with cryptorchidism and occupationally exposed humans working in the plastics industry. Therefore, it is vital to conduct more epidemiological and experimental studies to understand whether and how phthalates can induce the cancer of male reproductive organs. The expansion of the current knowledge concerning expression and intracellular mechanisms of phthalates’ effects on the male and female HPG system is necessary for the efficient prevention and treatment of their adverse influences on human and animal reproduction.

## Figures and Tables

**Figure 1 ijerph-17-06811-f001:**
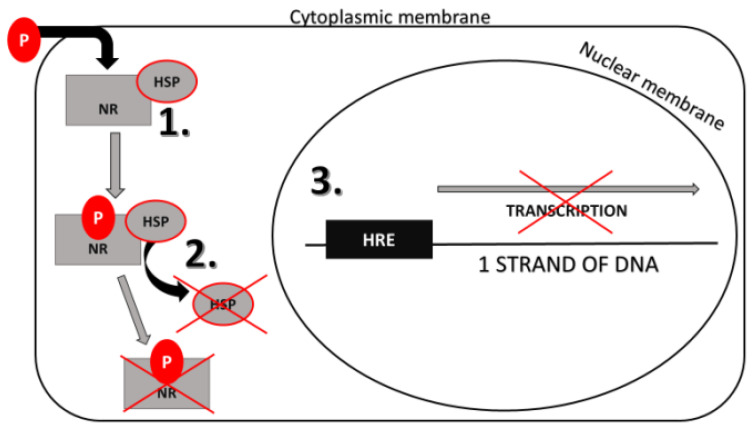
Phthalate action on a nuclear receptor: this figure shows an active form of antagonism when P blocks the conformational change of the complex with NR. 1. P binds to the NR; 2. HSPs cannot dissociate from the P-NR complex; 3. P-NR complex together with HSPs are in an inactive state and this inhibits transcription; HRE-hormone response elements; HSP-chaperones, NR-nuclear receptor, P-phthalate.

**Figure 2 ijerph-17-06811-f002:**
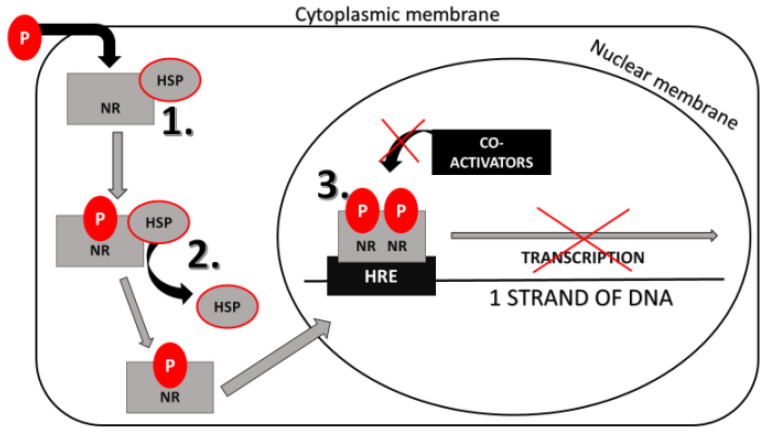
Phthalate action on nuclear receptors: this figure shows another form of antagonism when P prevents the NR from forming the correct conformation to activate the co-activators of transcription and thereby inhibit transcription. 1. P binds to the NR; 2. HSPs dissociate from the P-NR complex; 3. Co-activators cannot bind to the P-NR complex and therefore inhibit transcription; HRE-hormone response elements; HSP-chaperones, NR-nuclear receptor, P-phthalate.

**Figure 3 ijerph-17-06811-f003:**
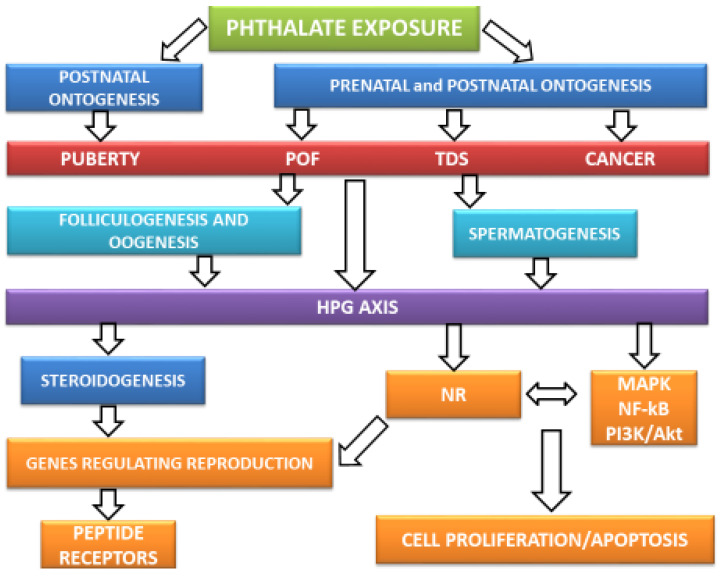
Side effects of phthalate action on reproductive health: prenatal and postnatal exposure to phthalates induces a wide spectrum of reproductive disorders. Phthalates can induce the puberty onset alternation and cancer occurrence in both females and males. In males, phthalates can induce testicular dysgenesis syndrome (TDS), which is connected with impaired spermatogenesis. In females, the exposure to phthalates can induce premature ovarian failure (POF), which is linked with impaired oogenesis and folliculogenesis. These reproductive disorders are mainly associated with a disrupted HPG axis that affects the process of steroidogenesis in both males as well as females. The activity of genes regulating reproduction can modulate steroidogenesis. These genes can be modulated by the activity of peptide and nuclear receptors (NR). Phthalates impair the peptide receptors and NR. Phthalates alter cell proliferation and apoptosis via crosstalk between MAPK, NF-kB, PI3K/Akt, and NR. This can lead to impaired spermatogenesis and cancer occurrence.

**Table 1 ijerph-17-06811-t001:** Examples of inducers and effectors of cell proliferation and apoptosis [43,48,49].

	Inducers	Effectors
**Cell Proliferation**	mitogens (growth factors—EGF, BDNF), survival factors (Bcl-2), steroids (testosterone)	cyclin-dependent kinases
**Apoptosis**	DNA damage, lack of nutrients, toxins, growth factors (TGF)	caspases, Ca2+ -dependent proteases (calpain)

**Table 2 ijerph-17-06811-t002:** Primary and secondary metabolites of the selected diesters of phthalates [66].

Phthalate Diester	Primary Metabolite	Secondary Metabolite
DMP	MMP	-
DEP	MEP	-
DnBP	MnBP	MHnBP
DiBP	MiBP	MHiBP
BBzP	MBzP	-
DEHP	MEHP	MEHHP
		MEOHP
		MECPP
DPeP	MPeP	-
DCHP	MCHP	-
DiDP	MiDP	MHiDP
		MOiDP
		MCiDP
DiNP	MiNP	MHiNP
		MOiNP
		MCiNP
DnOP	MnOP	MCPP

Legend: BBzP—benzylbutyl phthalate, DCHP—dicyclohexyl phthalate, DEP—diethyl phthalate, DEHP—di(2-ethylhexyl) phthalate, DMP—dimethyl phthalate, DiBP—di-iso-butyl phthalate, DiDP—di-iso-decyl phthalate, DiNP—di-iso-nonyl phthalate, DnBP—di-n-butyl phthalate, DnOP—di-n-octyl phthalate, DPeP—dipentyl phthalate, MBzP—monobenzyl phthalate, MCHP—monocyclohexyl phthalate, MCiDP—mono(carboxy-iso-decyl)phthalate, MCiNP—mono(carboxy-iso-decyl)phthalate, MCPP—mono-(3-carboxypropyl) phthalate, MECPP—mono-(2-ethyl-5-carboxypentyl) phthalate, MEHHP—mono(2-ethyl-5-hydroxyhexyl) phthalate, MEHP—mono-(2-ethylhexyl) phthalate, MEOHP—mono(2-ethyl-5-oxohexyl) phthalate, MEP—monoethyl phthalate, MHiBP—mono(2-hydroxy-iso-butyl)phthalate, MHiDP—mono(hydroxy-iso-decyl)phthalate, MHiNP—mono(hydroxy-iso-nonyl)phthalate, MHnBP—mono-(-3-hydroxy-n-butyl)phthalate, MiBP—mono-iso-butyl phthalate, MiDP—mono-iso-decyl phthalate, MiNP—mono-iso-nonyl phthalate, MMP—monomethyl phthalate, MnBP—mono-n-butyl phthalate, MnOP—mono-n-octyl phthalate, MOiDP—mono(oxo-iso-decyl)phthalate, MOiNP—mono(oxo-iso-nonyl)phthalate, MPeP—monopentyl phthalate.

**Table 3 ijerph-17-06811-t003:** Phthalates effect on hypothalamic–pituitary–gonadal (HPG) axis and steroidogenesis (in vivo, in vitro experiments).

**Females**					
**Phthalates**	**Dose Effect (mg/kg/day)**	**Animal/Cell Model**	**Time of Exposure**	**Effect**	**References**
DEHP	300	Sprague-Dawley rats	prenatal	↓ estradiol	Martinez-Arguelles et al. [173]
DEHP	1, 50, or 300	Sprague-Dawley rats	prenatal	↑ FSH	Meltzer et al. [138]
	300			↓ estradiol	
DEHP	300	Sprague-Dawley rats	postnatal	↓ pregnenolone, progesterone	Nuttall et al. [174]
DEHP	30	Wistar rats	prepubertal	↑ LH	Carbone et al. [175]
DEHP	1000, and 3000	Wistar rats	postnatal	↑ GnRH	Liu et al. [132]
DEHP	3000			↓ FSH, LH, estradiol, progesterone, testosterone	
DEHP	1000, and 500	Wistar rats	postnatal	↑ GnRH	Liu et al. [176]
DEHP	1, 10, 100 μg/mL	Mouse antral follicles (CD-1 mice)	postnatal	↓ progesterone, dehydroepiandrosterone, androstendione, testosterone, estradiol	Hannon et al. [121]
	1, 10, 100 μg/mL			↓ Cyp11a1	
	100 μg/mL			↓ Cyp17a1	
	10 μg/mL			↑ Cyp19a1, Hsd17b1	
	100 μg/mL			↓ Cyp19a1, Hsd17b1	
	100 μg/mL			↑ Hsd3b1	
DBP	0.01, 0.1 and 1000	CD-1 mice	postnatal	↓ estradiol, ↑ FSH	Sen, Liu and Craig [177]
	1000			↓ Star, Hsd3b	
	0.01, 0.1 and 1000			↓ P450scc, Cyp17a1, Hsd17b, Cyp19a1	
	0.01			↑ Star, Hsd17b	
DEHP	500 and 750	CD-1 mice	prenatal	↑ estradiol (F1 generation)	Brehm et al. [178]
	20			↑ estradiol (F3)	
	500			↓ testosterone (F1)	
	20			↓ testosterone (F2)	
	20 and 500			↓ testosterone (F3)	
	200			↓ progesterone (F2)	
	500			↓ FSH (F1)	
	500			↑ FSH (F3)	
	20			↑ LH (F1)	
**Males**					
**Phthalates**	**Dose Effect (mg/kg/day)**	**Animal**	**Time of Exposure**	**Effect**	**References**
DnBP	500	Sprague-Dawley outbred CD rats	prenatal	↓ Star, Cyp11a1, Cyp17a1	Thompson et al. [179]
DEHP	100, 300, 750	Sprague-Dawley rats	prenatal	↓ testosterone	Martinez-Arguelles et al. [173]
DEHP	3 and 30	Wistar rats	prepubertal	↑ GnRH	Carbone et al. [175]
DnBP	850	Sprague-Dawley rats	prenatal	↓ Cyp11a1, Hsd3b, Star	Zhu et al. [180]
DEHP	250, 500, or 750	Sprague-Dawley rats	postnatal	↓ testosterone, FSH, LH	Ha et al. [181]
DnBP	100 and 500	Wistar rats	postnatal	↓ Hsd17b, Hsd13b	Giribabu et al [182]
				↓ testosterone, FSH, LH	
DEHP	500 and 1500	Sprague-Dawley rats	postnatal	↓ GnRH	Qin et al. [183]
	100, 500, 1500			↑ Star, Hsd3b	
	1500			↑ Hsd17b	
DEHP	5 and 50 µg/kg/d	Long Evans rats	prenatal	↓ Hsd17b	Abdel-Maksoud, Ali and Akingbemi [184]
BBzP	10	Sprague-Dawley rats	pubertal	↑ testosterone	Lv et al. [185]
	1000			↓ testosterone	
	100			↑ Cyp11a1	
	10, 100 and 1000			↑ Hsd3b	
DEHP	500	Crl:CD rats	postnatal	↑ Cyp4a	Yamaguchi et al. [186]

**Legend:** BBzP—benzylbutyl phthalate, DBP—dibutyl phthalate, DEHP—di(2-ethylhexyl) phthalate, DiNP—di-iso-nonyl phthalate, DnBP—di-n-butyl phthalate, FSH—follicle-stimulating hormone, GnRH—gonadotropin-releasing hormone, LH—luteinizing hormone.

**Table 4 ijerph-17-06811-t004:** Phthalates effect on the HPG axis and steroidogenesis (epidemiological studies).

**Men**			
**Phthalate**	**Time of Exposure**	**Effect**	**References**
ΣDEHP	postnatal (more than 60 years)	↓ free testosterone, total testosterone	Woodward et al. [198]
ΣLMWP	postnatal (20–30 years)	↓ free testosterone, total testosterone	
MEHP	postnatal	↑ DHT, estradiol, P450AROM, SRD5A	Chang et al. [107]
MEHHP			
MEOHP			
MECPP			
MiBP	postnatal	↓ testosterone	Al-Saleh et al. [197]
MEHHP		↓ FSH	
MEP		↑ estradiol	
MEHP		↑ FSH, LH	
		↓ testosterone/LH, testosterone/estradiol	
MEHP	prenatal	↑ total testosterone (postnatal, 20 years old)	Hart et al. [193]
MiNP			
ΣDEHP			
ΣDiNP			
ΣHMWP			
ΣMEHP	postnatal (boys)	↓ testosterone	Wen et al. [196]
MBP	postnatal	↓ total testosterone, free testosterone, LH	Pan et al. [195]
MiBP			
MHiNP	postnatal	↑ FSH, LH	Axelsson et al. [194]
MOiNP		↑ FSH, LH	
MEHP	prenatal (boys)	↓ progesterone, INSL3, inhibin	Araki et al. [207]
ΣDEHP	postnatal	↓ testosterone	Specht et al. [208]
ΣDiNP			
MEHP	postnatal	↓ testosterone	Jurewicz et al. [80]
MEHP	postnatal	↓ testosterone/LH, testosterone/FSH, total testosterone, free testosterone, testosterone/estradiol	Joensen et al. [209]
MiNP		↓ testosterone/LH, testosterone/FSH, ↑SHBG	
ΣDEHP	postnatal	↓ testosterone, LH, FSH	Pan et al. [210]
ΣDBP			
MEHP	postnatal	↓ free testosterone, estradiol	Mendiola et al. [199]
MEHHP		↓ free testosterone, ↑ SHBG	
MEOHP		↓ free testosterone, ↑ SHBG	
MEP	postnatal	↓ LH	Jonsson et al. [203]
MBzP	postnatal	↓ FSH	Duty et al. [204]
**Women**			
**Phthalate**	**Time of Exposure**	**Effect**	**References**
MiBP	postnatal	↑FSH, ↓estradiol/FSH	Cao et al. [211]
MnBP		↓estradiol, ↑FSH, ↓estradiol/FSH	
MMP		↓estradiol, ↑FSH, ↓estradiol/FSH	
MEOHP		↑FSH	
MEHHP		↑FSH, ↓estradiol/FSH	
ΣLMWP		↓estradiol, ↓estradiol/FSH	
ΣHMWP		↓estradiol/FSH	
MHBP	postnatal	↑testosterone	Cathey et al. [205]
MEP		↓testosterone	
MEHHTP		↓ progesterone	
∑MEHP	prenatal	↓ progesterone	Wen et al. [212]
	postnatal (girls)		
MEHP	prenatal (girls and boys)	↓ testosterone/estradiol, progesterone, inhibin, INSL3	Araki et al. [207]
MEHP	prenatal	↓ free testosterone, free testosterone/estradiol (cord serum in newborns)	Lin et al. [213]
MEHHP			
ΣDEHP			

**Legend:** DHT—dihydrotestosterone, FSH—follicle-stimulating hormone, INSL3—insulin-like peptide 3, LH—luteinizing hormone, MBzP—monobenzyl phthalate, MCNP—monocarboxy-isononyl phthalate, MECPP—mono-(2-ethyl-5-carboxypentyl) phthalate, MEHHP—mono(2-ethyl-5-hydroxyhexyl) phthalate, MEHHTP—mono(2-ethyl-5-hydroxyhexyl) terephthalate, MEHP—mono-(2-ethylhexyl) phthalate, MEOHP—mono(2-ethyl-5-oxohexyl) phthalate, MEP—monoethyl phthalate, MHBP—mono(3-hydroxybutyl)phthalate, MiBP—mono-iso-butyl phthalate, SHBG—sex hormone-binding globulin, ΣDEHP—sum of di(2-ethylhexyl) phthalate metabolites, ΣDiNP—sum of di-iso-nonyl phthalate metabolites, ΣHMWP—sum of high-molecular weight phthalate metabolites, ΣLMWP—sum of low-molecular weight phthalate metabolites, ΣMEHP—sum of mono-(2-ethylhexyl) phthalate metabolites.

**Table 5 ijerph-17-06811-t005:** Estrogenic/anti-estrogenic and androgenic/anti-androgenic affinity of the selected phthalates [237].

	Estrogenic Affinity	Anti-Estrogenic Affinity	Androgenic Affinity	Anti-Androgenic Affinity
**DEP**	Yes	ND	ND	ND
**DnBP**	Yes	Yes	Yes	Yes
**DiBP**	Yes	ND	ND	Yes
**BBzP**	Yes	Yes	ND	Yes
**DiNP**	Yes	ND	ND	ND
**DEHP**	Yes	Yes	Yes	Yes
**DCHP**	Yes	Yes	ND	Yes

**Legend:** ND—no data.

**Table 6 ijerph-17-06811-t006:** The function of some genes regulating male and female reproductive system development [242].

**Gene**	**Function in Male Reproductive Development**
FGF9	proliferation and differentiation of Sertoli cells, formation of testicle tubules and Leydig cells
GATA4	triggers anti-Müllerian hormone secretion in Sertoli cells and regulates secretion of testosterone by Leydig cells
PTCH	expression is activated by Hh signaling pathway, which regulates the process of genital tubercle growth and differentiation in a masculine way [243]
SF1	anti-Müllerian hormone secretion in Sertoli cells and regulation of the secretion of testosterone by Leydig cells; secretion of insulin-like peptide 3
SOX9	differentiation of indifferent gonads to the testes; stimulation of anti-Müllerian hormone secretion in Sertoli cells
SRY	necessary gene for male sexual development
WT1	anti-Müllerian hormone secretion in Sertoli cells
**Gene**	**Function in Female Reproductive Development**
DAX1	testis development inhibition by acting antagonistically to SRY
FOXL2	ovarian development and function
RSPO1	positive regulation of WNT signaling pathway [244]
WNT4	ovary development; cell proliferation, apoptosis and differentiation within the female reproductive system

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
