# Peer review of "Effects and Mechanisms of Phthalates’ Action on Reproductive Processes and Reproductive Health: A Literature Review"

_ijerph, 2020, doi:10.3390/ijerph17186811_

Round 1

Reviewer 1 Report

The review is clearly divided, the tables are well-arranged
the list of abbreviations is complete.

1. Due to the topic of the manuscript, I expected more information about the relationship of phthalates to oogenesis.

2. Chapter 7.4. : Can the authors include any other studies on the effect of phtalates on oogenesis?

3. Figure 3: This figure should include not only the effect on spermatogenesis but also oogenesis and embryogenesis.

4. Can the authors explain what is the developmental window in the relation to endocrine disruption by phtalates?

Reviewer 2 Report

The review by Hlisníková H et al. describes the deleterious effect of Phthalates on reproductive health, highlighting the different molecular mechanism of action. The authors have shown a clear review and, in general, the paper is well-written and comprehensive. However, some criticism should be better analyzed.

1) The manuscript includes a lot of data that are compelling in their breadth, but the authors do little to interpret them. Instead, seems that these sections read like simple lists of previous results. The authors need to connect these data reporting also negative and controversial information.

2) Regarding “epigenetic processes” the authors should describe the main epigenetic changes mediated by phthalates that affects reproductive system, for example in section 2.4.3.

3) In Figure 1 and 2 it is not necessary to indicate “A” and “B”.

8) There are some typing error throughout the manuscript and in tables. Please proofread and fix. Please, focused also on standard English revision.

Reviewer 3 Report

The review constitutes important compendium of knowledge on phthalates action in reproductive system. It is based on the newest literature data. The manuscript is well, clearly and detailed described (there are references to e.g. mechanisms of action, effect on male/female including pregnancy, reproductive system development and function, cellular processes, proteins e.g. SHBG). Tables and schemes are very helpful. However some suggestion regarding to the information that can be add making the review broaden in terms of the basic knowledge are presented below

-For “2.4. Mechanisms of steroid action” it will be worth to add information on membrane androgen receptor in male reproductive system (e.g. Kaminska et al., Andrology; 2020)

-Non-receptorial action of EDCs should be mention too

-2.4.1 for this chapter please include clear information on ER types and PR types of receptors. Please consider to add information on estrogen-related receptors ERR

-Please refer to GPER expression and role in male and female reproductive system cells (e.g. Kotula-Balak et al., Cell Tissue Res 2018)

-Daily exposure dose (animals/human) and phthalates presence in the environment with their half-life and metabolism in organism need to be presented

(at 10-4-10-6mol/L)

- It will be nice to have division of phthalates on estrogenic/antyandrogenic affinities

Reviewer 4 Report

Overall I enjoyed reading this review. I find it an informative and almost complete summary of the knowledge around the effects on male and female reproduction of phthalates. Evidences are a bit unbalanced towars animal models, but I understand that available in vivo evidences on phthalates effects are fewer and not always of good quality. Some parts of the work, however, may require some clarifications before publication.

I particular, I would suggest the Authors to focus on these points:

Paragraph 2 represents a digression from the main topic (mechanisms of phthtalates...) towars physiological mechanisms of hormone signalling. Although some background may be necessary, the section may be too long. Also paragrahps 6 and 7 seem a good summary of phthalates molecular effects. I would suggest the Authors to summarize these paragraphs (mantaining the bulk of the last ones, that are main goal of the review)

Paragraph 3 - the concept and definition of EDs should be expanded, including the concept of "cocktail effect". Since in vivo effects of Phthalates may be exerted in synergy with other EDs (see for example PMID: 26544531 and PMID: 25244397)

Lines 241-242 - "instead"? please rephrase.

Lines 263-264 - please see also PMID: 26601918

Lines 313 -324 - this is true, but differences may arise not only from genetic differences, but also from their interaction with the environment and to (possibly) different EDs mixtures.

Lines 359-362 - These studies focused on PVC exposure. PVC structure  may possibly contain chroride compounds (PCBs...), phenols and phthalates. All of these may have endocrine disrupting properties.  This subsection seems to lose the focus on Phthalates.
